# Finding the Appropriate Therapeutic Strategy in Patients with Neuroendocrine Tumors of the Pancreas: Guideline Recommendations Meet the Clinical Reality

**DOI:** 10.3390/jcm10143023

**Published:** 2021-07-07

**Authors:** Sebastian Krug, Marko Damm, Jakob Garbe, Senta König, Rosa Lynn Schmitz, Patrick Michl, Jörg Schrader, Anja Rinke

**Affiliations:** 1Clinic for Internal Medicine I, Martin-Luther University Halle/Wittenberg, Ernst-Grube-Straße 40, D 06120 Halle, Germany; marko.damm@uk-halle.de (M.D.); jakob.garbe@uk-halle.de (J.G.); senta.koenig@gmail.com (S.K.); rosa.schmitz@uk-halle.de (R.L.S.); patrick.michl@uk-halle.de (P.M.); 2I. Medical Department, University Medical Center Hamburg-Eppendorf, Martinistrasse 52, D 20246 Hamburg, Germany; jschrader@uke.de; 3Department of Gastroenterology and Endocrinology, University Hospital Marburg, Baldinger Strasse, D 35043 Marburg, Germany; sprengea@uni-marburg.de

**Keywords:** NET, neuroendocrine, pancreas, survey, treatment, PRRT, guidelines

## Abstract

The systemic treatment of patients with pancreatic neuroendocrine tumors is based on placebo-controlled trials and long-established chemotherapy approaches. In addition, peptide receptor radionuclide therapy (PRRT) was approved as a parallel approach for pancreatic neuroendocrine tumors (NET), in addition to small bowel NET, after the NETTER-1 trial. The current ESMO and NCCN guidelines attempted to describe treatment algorithms for pancreatic NET based on the current data. In our survey, we recorded therapy decisions for the first- until the third-line of therapy in German-speaking countries (Germany, Austria, and Switzerland) using fictional case reports and discussed these in the context of the current ESMO guidelines. Compared with the recommendations of the guidelines, PRRT was used more frequently and earlier. In patients with NET G1/G2 Ki-67 < 10%, the therapy algorithm consisting of somatostatin analogs (SSA)-PRRT-targeted therapy is a relevant approach. In clinical situations where chemotherapy is primarily used (remission pressure, Ki-67 > 10%), second-line PRRT was found acceptance and was often considered prior to targeted therapies. Despite the lack of prospective controlled trials, our study demonstrated the pivotal impact of PRRT. Therefore, further studies should compare PRRT with chemotherapy in pancreatic NETs in different clinical settings in first- and second-line approaches.

## 1. Introduction

Gastroentero-pancreatic neuroendocrine tumors (GEP-NETs) comprise a broad clinical and therapeutic spectrum of malignancies. The medical and socioeconomic relevance has increased significantly due to the rising incidence and the long follow-up in the curative, as well as long therapeutic care, in the palliative setting [1]. Within the last decade, the spectrum of therapies has expanded significantly. In addition to the already established therapies of somatostatin analogs (SSA), locoregional liver-specific therapies such as TACE/SIRT, and chemotherapy for pancreatic NET, the targeted therapies everolimus/sunitinib have been approved for pancreatic and peptide radioreceptor therapy (PRRT) for all somatostatin receptor-positive GEP-NETs [2,3,4,5].

Either a placebo arm or a double standard dose SSA served as the comparator in the more recent pivotal trials. The NETTER-1 trial of PRRT in midgut NETs also introduced a treatment protocol that both defined a fixed number of cycles and radiation doses and established the continuous use of SSA during therapy [5]. In addition to the numerous national guidelines for neuroendocrine neoplasms [6,7,8], the American (NCCN) and European (ENETS and ESMO) societies have launched the current guidelines that define comprehensive recommendations for the various therapeutic approaches [9,10,11].

As predictive markers for the treatment choices are not available, the recommendations for systemic therapies are based on the proliferative capacity of the disease (measured in Ki-67), the extent of metastasis, tumor-related symptoms, functionality, and patient-specific comorbidities [10]. While there is a well-established treatment algorithm consisting of SSA-PRRT-targeted therapy (+/−locoregional therapy) for patients with small bowel NET, the situation in pancreatic NETs is more complex and more heterogeneous [11].

Based on ancient studies, the 30-year established streptozotocin-containing (STZ) combination chemotherapy is faced with a variety of other options, including SSA, targeted therapies, PRRT, and a temozolomide/capecitabine regimen. Therefore, an evidence-based definite therapeutic algorithm supported by comparative randomized studies in pancreatic NET does not exist. The extent to which patient preference additionally matters for determining a therapy in pancreatic NET can also not be precisely determined.

In the present study, we used a survey to inquire about feasible therapeutic strategies based on fictional case reports of pancreatic NETs. The aim of the survey was to evaluate the use of the proposed therapeutic options and to develop a therapeutic algorithm in comparison to the existing guidelines.

## 2. Methods

Ethical approval was obtained from the local ethical review committee of Martin-Luther-University Halle/Wittenberg (number: 2020-093; June 2020). A total of 32 questions were developed, including four general questions regarding the characteristics of the physicians and facilities, 14 case presentations, and a respective reasoning question (overall, 28 questions) to evaluate the management of pancreatic NET (PanNET) patients. Either several answers of a given choice, only one answer (multiple choice), or yes/no answers were possible. In some cases, free text was possible to specify the answer.

The survey was performed between 1 July and 31 December 2020 in the following three German-speaking countries: Austria, Switzerland, and Germany. The survey was open to physicians who treated NET patients in inpatient and outpatient settings and was distributed via the German NET-Registry and via personal contact. In Appendix A the fictional case reports are summarized. The full survey is provided in the Appendix A.

The software LimeSurvey version 3.0 (LimeSurvey GmbH, Berlin, Germany) was used to conduct the online survey. Descriptive statistics were calculated using Office Excel 2016 (Microsoft Corporation, Redmond, WA, USA) and GraphPad Prism 5 (GraphPad Software, San Diego, CA, USA).

## 3. Results

### 3.1. Characteristics of Physicians and Facilities

A total of 98 participants took part in the survey, and 75.5% (*n* = 74) of the participants answered all the questions completely. Most physicians worked in a university hospital (71.4%, *n* = 70), followed by non-university hospitals (16.3%, *n* = 16) and specialist practices (7.2%, *n* = 7). The question of clinical specialization was answered as follows: oncology (19.4%, *n* = 19), endocrinology (21.4%, *n* = 21), gastroenterology (13.3%, *n* = 13), surgery (11.2%, *n* = 11), nuclear medicine (23.5%, *n* = 23), and internists (2%, *n* = 2) (Figure 1, left column).

Five colleagues reported holding multiple specialty qualifications (5.1%, *n* = 5). In most facilities, 11–25 pancreatic NET (PanNET) patients were treated annually (31.6%, *n* = 31). Subsequently, NET centers with more than 50 cases per year appeared (27.6%, *n* = 27), and finally, centers with 26–50 cases per year followed (17.4%, *n* = 17) (Figure 1, middle column). Typically, the physicians spent 5–20% of their time treating NET patients (53%, *n* = 52). Only seven participants spent more than 40% of their time on the care of NET patients (Figure 1, right column).

### 3.2. Concept of Survey Evaluation

A total of 14 clinical scenarios were presented to the participants. Using the current ESMO guidelines on gastroentero-pancreatic neuroendocrine neoplasms (GEP-NENs), the cases were allocated to a treatment strategy and line of therapy for PanNETs (Figure 2). Two cases addressed the feasible curative concepts (C10–11) and served as internal validation that primarily well-informed NET specialists participated in the survey. For the first-line therapy in PanNETs, four questions were asked, including four NET G2 patients (C2, 6, 8, and 14). A total of six case reports were presented for the systemic second-line therapy, including three for NET G1 (C1, 3, and 5); two for NET G2 (C7 and 12); and one for a PanNET G3 (C13). Finally, there were two cases that evaluated the impact of the third-line therapy (C4 and C9). The results of the participating physicians are presented in Figure 3. The results of the survey for the corresponding disciplines are attached as Appendix A.

### 3.3. Case Presentations for First-Line Therapy

When asked about first-line therapy initiation, asymptomatic patients and Ki-67 < 10% revealed a priority use of SSA therapy (74.7%, *n* = 62), whereas “watch and wait” (6%, *n* = 5) and PRRT (6%, *n* = 5) were infrequently applied therapies in this situation. In clinical settings, where a PanNET G2 had growth dynamics with symptomatic metastases, the majority of doctors were in favor of initiating chemotherapy (64.6%, *n* = 51), followed by SSA treatment (12.7%, *n* = 10) and PRRT (7.6%, *n* = 6).

If liver metastases were present exclusively, and the primary tumor was located in the tail of the pancreas, a surgical procedure (23.7%, *n* = 18) was increasingly considered appropriate in addition to the use of chemotherapy (57.9%, *n* = 44). Otherwise, 9.2% (*n* = 7) of respondents also considered PRRT as a feasible option. If classical cytotoxic therapy (NET G2, Ki-67 15%) was refused by the patient and therapeutic remission was required, then PRRT (71.6%; *n* = 53) was preferred over SSA therapy (16.2%, *n* = 12) and targeted drugs (10.8%, *n* = 8).

### 3.4. Case Presentations for Second-Line Therapy

After progression under SSA therapy (NET G1 or NET G2 Ki-67 < 10%), PRRT was preferentially used if the disease was sufficiently controlled over a long time prior (54.3%, *n* = 44). A subset of participants also continued SSA therapy (14.8%, *n* = 12) or pursued locoregional therapy (22.2%, *n* = 18) in the case of isolated hepatic involvement. If tumor progression was rapidly observed under SSA (disease stabilization up to 12 months), then PRRT (43.4%, *n* = 36) was still primarily recommended; however, chemotherapy (32.5%, *n* = 27) and fewer tyrosine-kinase inhibitors (TKIs) (18.1%, *n* = 15) were also increasingly considered. If additional patient-related comorbidities occurred, the value of chemotherapy (12.1%, *n* = 12) decreased significantly, and PRRT was the most frequently selected (61.5%, *n* = 56). TKI therapy (17.6%, *n* = 16) was then prescribed the second-most frequently after progression under SSA.

In two case reports of PanNET patients in the higher G2 setting (Ki-67 15% and 20%), questions were asked about the subsequent therapies after failure under chemotherapy with STZ/5-FU or temozolomide and capecitabine (TEM/CAP). The difference between the two cases was only the type of metastasis (diffuse metastasis versus hepatic metastasis alone). In the first case, the use of PRRT (48.7%, *n* = 38) was favored over a change in chemotherapy (20.5%, *n* = 16) or a switch to targeted therapies (19.2%, *n* = 15). In the latter case with isolated hepatic metastasis, locoregional therapy (28.4%, *n* = 21) also received more consensus, and again, PRRT (37.8%, *n* = 28) was most often recommended before a change in chemotherapy (23%, *n* = 17).

In a clinical scenario characterized by a NET G3 (Ki-67 35%), where the initiated systemic first-line therapy with TEM/CAP had to be discontinued due to side effects, patient preference, and the documented tumor progress, most participants supported a switch to PRRT (41.9%, *n* = 31). Secondly, a shift to STZ/5-FU was recommended in 32.4% (*n* = 24), and, for some, the use of platinum-containing therapy with carboplatin/etoposide (18.9%, *n* = 14)) was also considered.

### 3.5. Case Presentations for Third-Line Therapy

For the third-line therapy, two cases were introduced in which patients with metastatic PanNET (G2 Ki-67 < 10%) received chemotherapy or PRRT according to the standard of care after SSA failure. The patient under chemotherapy was progressive after six cycles of STZ/5-FU (over nine months) in the staging four months thereafter without treatment. The majority of participants opted for a therapeutic change in preferring PRRT (38.6%, *n* = 32) before a reinduction with STZ/5-FU (26.5%, *n* = 22). Therapy with TKIs or the alkylating agent dacarbacin (DTIC) was used in 10.8% (*n* = 9) and 8.4% (*n* = 7) of the cases. In the other case, PRRT (four cycles) was performed, under which a progressive disease occurred after 12 months. Here, TKIs (53.3%, *n* = 40) and chemotherapy (36%, *n* = 27) were mainly considered as further therapeutic options.

## 4. Discussion

The treatment of patients with GEP-NETs requires an interdisciplinary team with high expertise in diagnostic and therapeutic management. Since the incidence of GEP-NETs is much higher than mortality, the prevalence is steadily increasing, and the care of these patients requires several resources. In our survey, we demonstrated that the treatment of patients with pancreatic NETs in routine clinical practice differs from the recommended therapies in the guidelines. In this context, PRRT has a higher value than attributed to it by the guidelines, while targeted therapies and chemotherapy have become less relevant, particularly in subsequent therapeutic lines. The ESMO guidelines of 2020 were used as our reference for this purpose.

### 4.1. First-Line Therapy

When a patient with advanced and metastatic pancreatic NET is initially diagnosed, the Ki-67 index and accompanying symptoms are often used to select between SSA and chemotherapy [11]. In our study, Ki-67 (cut-off 10%), tumor-related symptoms, and documented tumor progression without prior therapy also proved to be valid criteria for SSA or chemotherapy. Beyond the subjectively evaluated tumor-related symptoms, the Ki-67 threshold of 10%, for which SSA therapy is approved in pancreatic NET, was implemented and arbitrarily chosen in the CLARINET study, and there were retrospective data favoring this cut-off [12,13]. However, there was only limited biological tumor evidence and rationale for this threshold. Interestingly, a recent study demonstrated the antiproliferative efficacy of SSA, particularly in NET G2 Ki-67 >10%, with a median PFS of 12.4 months [14].

Which chemotherapy regimen the respondents used was not explicitly asked in our survey. In addition to the established therapy concept of streptozotocin (STZ) and 5-FU according to the Uppsala or Moertel schedule [15,16], the current study by Kunz is prospectively investigating the value of temozolomide and capecitabine (TEM/CAP), with promising results thus far [17]. In clinical practice, oral combination therapy is also gaining popularity due to its simplicity of use [18]. However, STZ-based chemotherapy has been approved by the European Medicines Agency (EMA) since 2018, while TEM/CAP remains unapproved so far.

If there is exclusively liver metastasis of a NET G2 (Ki-67 15%) and the primary tumor is in the tail of the pancreas, then locoregional therapy procedures and removal of the primary are increasingly being considered. This approach is not discussed in the current ESMO guidelines but is mentioned as an option in the ENETS recommendations of 2016, including the possibility of using systemic therapies at a later stage [10]. The question of primary tumor resection in the presence of nonresectable metastases remains controversial and should be very carefully considered [19,20]. Thus far, there are only retrospective analyses on this issue.

If the patient, in a situation where SSA had no approval, refused classical chemotherapy, PRRT was recommended in three-quarters of the cases. This decision process is very interesting in that there are no published studies recommending the use of PRRT in pancreatic NET as the first systemic therapeutic option. In the NETTER-1 trial, where only midgut NET were included, prior therapy with SSA and tumor progression were required as inclusion criteria [5]. Both the COMPETE and OCCLURANDOM trials are testing PRRT against targeted therapy in the second line of therapy. However, conclusions regarding the effectiveness of PRRT in the first line of therapy can be drawn from the large unicentric retrospective PRRT evaluations by Baum and Imhof [21,22]. Here, PRRT was used as the first systemic therapeutic option in 11.4% and in 57.4%.

That patient preference relevance was presented in the PIANO study [23]. The major attributes for therapy choice were the overall survival; response to treatment; stabilization of tumor growth; and side effects, such as nausea, vomiting, and diarrhea. Therefore, the therapy modality of PRRT, which is perceived as elegant and with few side effects for patients, could be preferred [24]. Despite an overall good tolerability of PRRT, the rare and very relevant side effects and the long-term risk of myelodysplastic syndrome or acute leukemia should not be ignored and have been reported in 0.9–2% of cases [5,25,26].

### 4.2. Second-Line Therapy

After progression under SSA therapy in patients with NET G1/G2 Ki-67 < 10%, the majority opted for sequential therapy with PRRT irrespective of the period of disease stabilization under SSA. Chemotherapy and targeted therapies were less favored. The SSA-PRRT sequence has not been evaluated for pancreatic NET to date. However, for small bowel NET, this sequence is used as a common pathway. That targeted therapies are sparsely favored does not reflect the data for this class of agents. In both the RADIANT-3 and sunitinib trials, between 40 and 50% of patients were pretreated with SSA [2,3].

Despite this, targeted therapy is recommended in the guidelines after SSA with or without prior chemotherapy, even though disease stabilization is mainly achieved in approximately 11 months.

In patients with NET G2 in the higher proliferative range after chemotherapy failure (STZ/5-FU or TEM/CAP), PRRT was again recommended in most cases. Although this approach is not concordant with the guideline recommendations, it is comprehensible from a clinical point of view. As chemotherapy is preferentially used to induce remission in patients with a high tumor burden and tumor-related symptoms, the treatment response after progression is certainly a relevant consideration in the choice of therapy.

As an alternative to PRRT, a switch in the chemotherapy regimen has been recommended even prior to the use of targeted therapies. Dacarbazine (DTIC) therapy comes into consideration as a potential alternative for this purpose. DTIC was still able to achieve a median progression-free survival (mPFS) of 10 months in heavily pretreated patients [27]. In a small retrospective analysis of 28 patients, no significant difference in the mPFS was detected whether DTIC was used before or after STZ/5-FU [28]. Sequencing STZ-5-FU on TEM/CAP or vice versa cannot be recommended due to the lack of data.

### 4.3. Third-Line Therapy

In third-line therapy, an algorithm was presented after tumor progression under SSA-PRRT, as well as SSA-CTx (after 4 months off). In the first case, the majority preferred the use of targeted therapies prior to chemotherapy. This concept is surprising for pancreatic NET and would be in line with the treatment algorithm of small bowel NETs (SSA-PRRT-everolimus). In the SEQTOR trial, the sequence of targeted therapy and chemotherapy was indeed studied in patients with progressive disease, e.g., after SSA treatment. Whether this can be translated into a promising therapy sequence remains to be determined. In the second case, PRRT was considered before chemotherapy reinduction after the failure of an SSA-CTx sequence.

In this setting, targeted therapy was ascribed only a minor role. For both situations, arguments cannot be formulated based on existing evidence. Rather, the physician assessment, treatment tolerability, and treatment pressure affected the decision process. Recently published multicenter observational studies have investigated the sequencing strategies in well-differentiated neuroendocrine tumors. The Italian ELIOS study group found an advantage in the SSA-PRRT sequence mainly based on a good tolerability and quality of life [29]. In the Swiss NET registry study, no clear pattern emerged in the treatment sequences for pancreatic NET, with chemotherapeutic treatment used more frequently than PRRT in pancreatic NETs [30]. The same held true for the results from the German NET Registry [31].

## 5. Limitations

There are certain limitations to discuss in the context of our study. Although the survey had a well-balanced representation of the different disciplines, the survey of individual physicians did not reflect the decision-making process in a multidisciplinary tumor board (MDT). However, given the lack of robust evidence-based data for many clinical situations in pancreatic NET, deviations from the guideline recommendations are likely more common in NET compared to other diseases. In addition, the fictional cases may not reflect the complexity of real patient cases.

The argument that too many nuclear medicine physicians participated in the survey cannot be shared. Oncologists, endocrinologists, and nuclear medicine physicians each participated at approximately 20%. The spectrum of recommended therapies likely varies depending on the expertise of the physician; therefore, oncologists, endocrinologists, and gastroenterologists may be more diversified. As only a few participants belonged to the specialist practitioner sector (<10%), and no precise distinctions between the university and non-university sectors could be extracted from our survey.

To what extent the COVID-19 pandemic has had an impact on these decisions can only be speculated. Recommendations and our own studies indicate that chemotherapy, for example, is considered to have a higher risk of developing a severe COVID-19 infection compared with PRRT [32]. However, whether the COVID-19 pandemic has increased the use of PRRT appears unlikely, as PRRT is mainly performed at university centers, which have had to concentrate their resources due to the pandemic.

## 6. Conclusions

This survey provides a perspective on physician’s choices for therapy algorithms in patients with advanced or metastatic pancreatic NET based on fictional cases and subjective evaluations. Although randomized and prospective studies on PRRT in pancreatic NET are still in progress, PRRT is currently more popular in practice than is considered the norm. Based on our data, we created a practical, relevant therapy algorithm for pancreatic NET G1/G2 (Figure 4). The insights from Germany, Austria, and Switzerland, therefore, have to be evaluated in a prospective observational study to create an optimal management algorithm on the basis of a large database from multiple European centers.

## Figures and Tables

**Figure 1 jcm-10-03023-f001:**
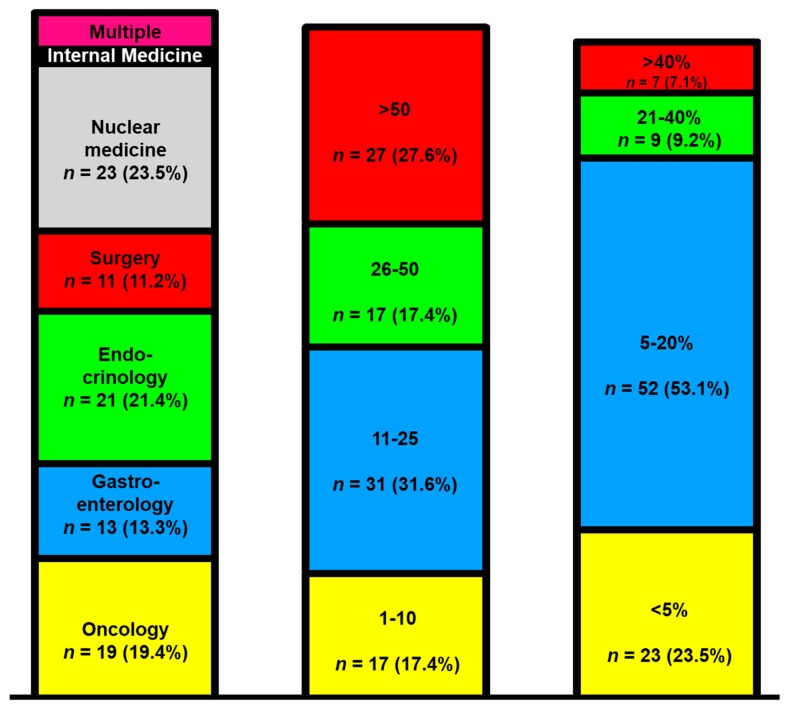
The survey responder disciplines (**left** column), treated pancreatic neuroendocrine tumor (NET) patients per year (**middle** column), and the time invested in NET care (**right** column).

**Figure 2 jcm-10-03023-f002:**
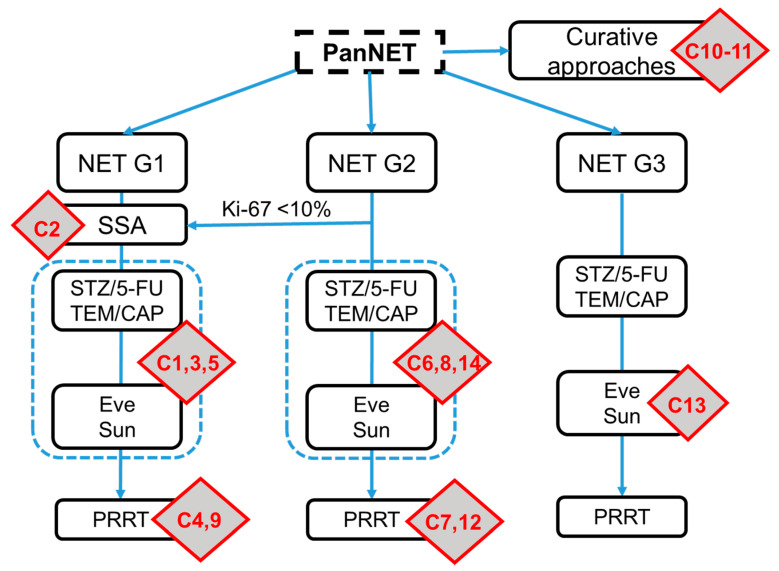
Allocation of the case reports to the current ESMO guidelines on gastroentero-pancreatic (GEP)-NETs. The C refers to the corresponding question in the survey and the red boxes illustrate the position of the questions within the ESMO therapy schema. The scheme was adapted based on the original version of the ESMO guidelines. SSA, somatostatin analogs; Eve, everolimus; Sun, sunitinib.

**Figure 3 jcm-10-03023-f003:**
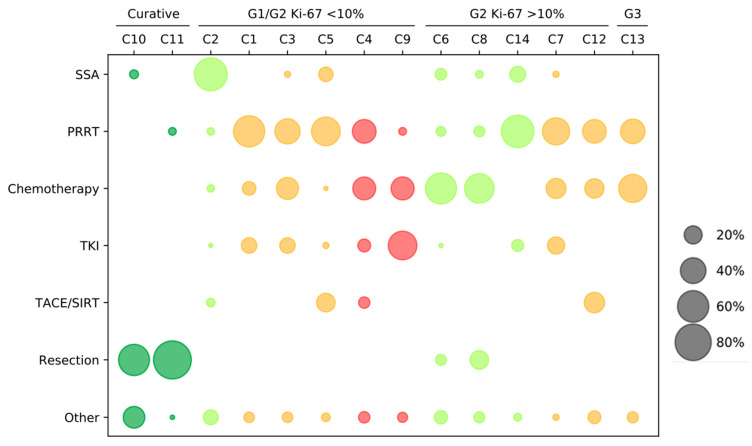
Treatment strategies in pancreatic NET (PanNET) cases. The selection of therapy based on the scenarios: curative, NET G1/G2 Ki-67 < 10%, NET G2 Ki-67 > 10%, and NET G3. The questions are grouped by first- to third-line therapies: the first-line questions C2, C6, C8, and C14 are marked in green; the second-line questions C1, C3, C5, C7, C12, and C13 are marked in orange; and the third-line questions C4 and C9 are marked in red. The size of the points reflects the percentage per case report. SSA = somatostatin analogs; PRRT = peptide receptor radionuclide therapy; tyrosine-kinase inhibitors (TKI) = everolimus and sunitinib; Chemotherapy = 5-FU/streptozotocin-containing (STZ) or temozolomide and capecitabine; and Other = watch and wait, combined approaches, or other not specified.

**Figure 4 jcm-10-03023-f004:**
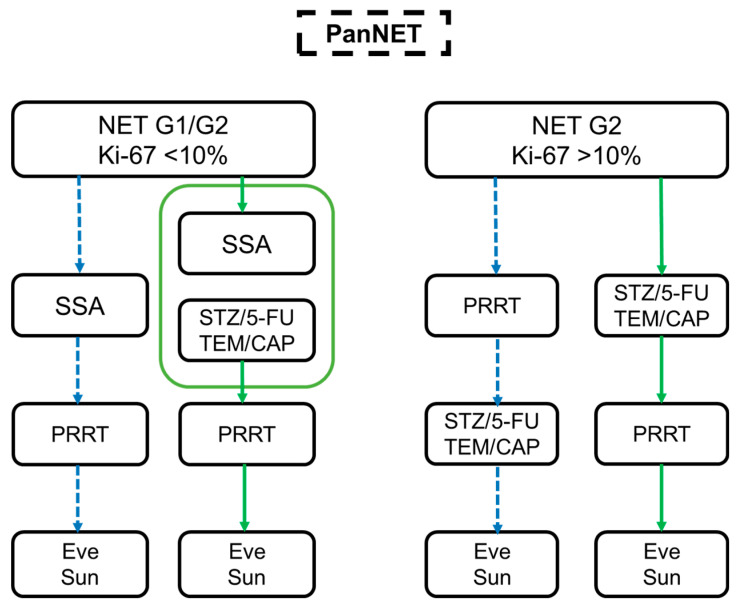
Summary of the survey. Treatment algorithm in pancreatic NETs based on different case scenarios for NET G1/G2 Ki-67 < 10% (**left** side) and NET G2 Ki-67 > 10% (**right** side). The dotted blue arrows represent the use of PRRT before CTx and TKI therapy (Eve and Sun), while the green arrows indicate the PRRT after SSA/CTx.

## Data Availability

The raw data presented in this study are available on request from the corresponding author.

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
