# Peer review of "Finding the Appropriate Therapeutic Strategy in Patients with Neuroendocrine Tumors of the Pancreas: Guideline Recommendations Meet the Clinical Reality"

_jcm, 2021, doi:10.3390/jcm10143023_

Round 1

Reviewer 1 Report

Dear Authors,

Overall an interesting MS with interesting final algorithm. The presentation of the data itself in the accompanying Figures especially Fig. 3 difficult to interpret without figures.

Detailed comments and questions to the paper are given below.

Page 2

Introduction

Line 60 It is not true what the authors suggest regarding the treatment of advanced pancreatic NETs that there are no definitive algorithms for the management of advanced pancreatic NETs. There are currently at least 4 guidelines for the management and further treatment of advanced pancreatic NETs, including the ESMO publication cited by the authors, followed by the ENETS, NANETS, and NCCN recommendations with the latest upgrade 03.2021. in the PanNET-7 section.

It would be more appropriate to emphasize the complexity of the therapeutic algorithm in pancreas.

Anyway, the authors build their whole paper on the basis of ESMO recommendations.

Line 61 here agree indeed preference in treatment selection based on analysis of patient's condition, NET type, advancement of comorbidities etc. is not precisely defined.

Methods

Line73 The presentation of clinical cases in the supplemental materials is very general and does not take into account the complete medical history of the patients, their comorbidities, depending on many elements included in the medical history, we decide on the treatment, which unfortunately was not taken into account in the presented cases on the basis of the clinical information provided. 

Page 3

Fig1.

WIll be very helpful if authors used numbers and (%) as presented data sets in bars.

Page 4.

Fig. 2 The allocation of the case reports and the numbering of the included questions within the ESMO algorithm (it should be noted that it is modified from the original ESMO scheme) seems somewhat random. The question to the authors is whether it is possible to have a more structured system of asking questions corresponding to the ESMO scheme, e.g. depending on the degree of G-differentiation and clinical initial stage in order to facilitate the analysis of the presented results on the basis of Fig. 3.

Page 5.

Fig 3. Presentation of survey results on the basis of graphical presentation is difficult to interpret the results and its comprehensive understanding. Determining the answers on the basis of the diameter of the circles without specifying their values is unclear. As a whole the results presented in such a way are not very clear for the reader who has no insight what the symbols (circles) mean and what their numerical value is.

My suggestion is that the authors try to use the Sankey diagram of sequential treatment in a group of patients with advanced NET in response to the varied clinical situation presented in their questionnaire. Which may be clearer in interpreting the results.

Page 6

Discussion

Line 187

In our survey, we demonstrated that the treatment of patients with pancreatic NETs in routine clinical practice differs significantly from the recommended therapies in the guidelines.

Where the authors cite statistical results showing significant differences in outcomes based on their study.

Page 8

Limitation

Line292 fully agree with this statement.

Conclusion

Line 308

This study provides the first real-world perspective on the choice of treatment algorithms by physicians in patients with advanced or metastatic pancreatic NET.

This is not an analysis of a real-world perspective of treatment algorithm choice, only a hypothetical treatment regimen based on exemplary (fictional) patients and treatment choices by physicians participating in the survey, this should be accurately described.

The description in the conclusion as a real-world perspective on management algorithm selection would have to be based on actual data from a prospective clinical trial or a meta-analysis type study.

Line 311

The management algorithm developed by the authors with the earlier use of PRRT, now a therapy of this type is called Radioligand Therapy (RLT) is highly consistent with recent research and the experience of centers caring for NET patients.

Line 312

The conclusions drawn by the authors do not so much need to be re-evaluated from the European perspective of centers outside Germany, Austria, or Switzerland, but should be integrated in a retrospective evaluation of actual data based on treated patients and applied management algorithms and the creation of a potential multicenter prospective study followed by the construction of an optimal management algorithm on the basis of a large database from multiple European centers.

Reviewer 2 Report

Krug et al present a very interesting survey among NET specialists with regard to preferences for the various treatment options in a number of fictional cases. The results show that PRRT is more favored in reality than in the current guidelines.

I have one minor comment. I think it would be interesting to show the results as shown in fig 3 for the largest categories of specialists. My estimate would be there will be different prefences when asked to oncologists versus endocrinologists for example, which may be intersting to show because it could underline the importance of the MDT.
